# MATEVOLVE: A SYNERGISTIC SYMBOLIC–LLM AGENT FOR MULTI-OBJECTIVE MATERIALS DESIGN

## ABSTRACT

The design of novel materials is fundamentally constrained by the immense chemical space, which renders traditional **enumeration-screening** methodology computationally prohibitive and inefficient. This paper introduces a paradigm shift towards **insight-exploration-validation**, enabling an intelligent and evolutionary exploration of material design pathways. We actualize this paradigm through **MatEvolve**, a synergistic symbolic–LLM agent that reconceptualizes material design as a closed-loop, programmatic evolution task. Central to MatEvolve is a novel symbolic formalism, Material Edit Language, which empowers the agent to programmatically take chemical operations. The exploration trajectory is directed by a multifaceted guidance strategy, comprising a dynamic knowledge injection mechanism and a two-stage exploration strategy that balances broad exploration and deep optimization. Furthermore, a multi-objective fitness landscape ensures directional and efficient navigational guidance. These integrated strategies contribute to a **32.2%** improvement over direct material structure modification. Crucially, comparisons demonstrate that our insight-exploration-validation paradigm outperforms the traditional enumeration-screening approach by **33.6%**, highlighting its superior efficacy in navigating vast design spaces.

## 1 INTRODUCTION

The design and application of materials have always been a core driving force for the advancement of human civilization, positioning materials science as a cornerstone. Traditional materials research, which relies heavily on trial-and-error wet-lab experiments, is flawed by long cycles and high costs. Recently, deep learning has catalyzed a paradigm shift. The development of tools such as universal potential function prediction models has significantly accelerated materials screening, leading to a **enumeration-screening** paradigm. Merchant et al. (2023), based on graph neural networks, has efficiently predicted material stability, identifying millions of potential new crystals. Zeni et al. (2023) has enabled the de novo design of new materials with specific symmetries and chemical compositions through diffusion models, while Yang et al. (2024) provides a powerful tool for energy prediction with conditionally near-first-principles accuracy.

Despite its successes, the enumeration-screening paradigm faces three core limitations. First, the combinatorial explosion of elements, sites, and compositions creates a vast chemical space, which leads to algorithms getting stuck in local optima and poses a trade-off between predictive accuracy and exploration completeness. Second, its open-loop, funnel-like process lacks dynamic feedback for strategy adjustment, which allows errors to accumulate and prevents the system from correcting its exploration direction. Third, its reliance on scattered knowledge and expert heuristics restricts its generalization ability across different material systems. In contrast, work like Novikov et al. (2025) demonstrates that large language model (LLM) gudied evolutionary approaches can solve complex optimization problems, offering a new methodological blueprint for materials design.

In this paper, we propose a novel closed-loop paradigm: **insight-exploration-validation**. Unlike **static** enumeration-screening, as shown in Fig. 1, this paradigm injects symbolic insights into an LLM-based agent to conduct a more efficient and chemically intuitive exploration. Each exploration step is instantly evaluated by multi-objective performance metrics, and the results serve as feedback to guide the agent's subsequent decisions. This creates a closed-loop optimization mechanism, enabling the exploration process to **dynamically** converge on high-performance regions. To

implement this paradigm, we develop a LLM-friendly symbolic system, Material Editing Language (MEL), which maps comprehensive atom-level operations into code sequences that the LLM-based agent can fully parse and operate.

Building on this, we develop MatEvolve, a synergistic symbolic–LLM agent framework for materials design. MatEvolve consists of two core components: Material Editing Base (MEB) and Material Evolution Engine (MEE). MEB is a structured expert knowledge base which is constructed through an automated pipeline where LLM extracts material modification strategies from high-impact literature and converts them into the symbolic MEL format. During the design process, MatEvolve dynamically selects relevant knowledge from MEB as insights to guide the agent's optimization pathways. MEE executes a insight–exploration–validation closed-loop. To efficiently navigate in the vast chemical space, the engine employs a two-stage exploration strategy, beginning with a breadth-first exploration to cover diverse chemical spaces, and then shifting to a depth-first exploration to accelerate convergence. Newly generated material candidates are instantly evaluated by surrogate models across general, multidimensional performance dimensions. The evaluation results are integrated as feedback to continuously optimize MatEvolve's exploration strategy.

Applied to solid-state electrolytes and electrode materials, MatEvolve not only reproduces and extends known optimization pathways (Zhou et al., 2019) which can be found in Appendix. F, but also uncovers chemically plausible, interpretable candidates at a fraction of the computational cost. More importantly, as a generalizable framework, it offers a methodological blueprint for LLM-guided, dynamic, closed-loop materials design beyond current expert-guided, static, open-loop enumeration-screening approaches.

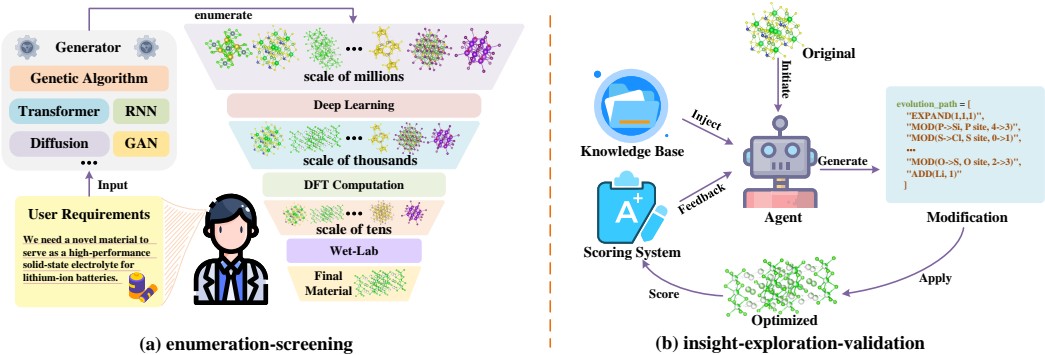

(a) enumeration-screening      (b) insight-exploration-validation

Figure 1: **Comparison of two paradiams**: (a) Enumeration-screening begins by generating millions of candidate molecules and then uses a funnel-like process, from fast deep learning models to precise DFT computations and finally wet-lab experiments, to filter them and identify the best material. (b) Insight-evaluation-validation starts with initial molecules and uses agents to apply targeted modifications. Guided by a knowledge base and a scoring system, the agent continuously refines materials in a feedback loop, evolving them towards a superior material.

In summary, the main contributions of this study are as follows:

- We propose an insight-exploration-validation paradigm and its accompanying MEL, which overcome the expert-guided, static, open-loop limitations of the enumeration-screening paradigm.

- We develop the MatEvolve framework and significantly enhance the efficiency and success rate through dynamic knowledge injection from MEB and a two-stage exploration strategy embedded in MEE which balances the breadth and depth of material design.

- Our experiments demonstrate the broad applicability and high performance of MatEvolve in designing solid-state electrolytes and electrode materials, achieving a performance improvement of 32.2% over direct modification of material structure and exhibiting 33.6% greater efficacy than enumeration-screening methods.

## 2 RELATED WORKS

### 2.1 MATERIALS DESIGN

Computational methods have advanced materials discovery, yet often rely on linear, open-loop workflows. For instance, Zhang et al. (2019) pioneered an unsupervised learning scheme using modified X-ray diffraction patterns to screen for solid-state ion conductors like LLZO and LGPS in data-scarce scenarios. Subsequently, Choi et al. (2021) employed active learning to enhance the accuracy of machine learning models for predicting the mechanical properties of solid-state electrolytes. More recently, Chen et al. (2024) demonstrated a massive high-throughput screening pipeline, culminating in the experimental synthesis of $Na_2LiYCl_6$ and successfully closing the prediction-synthesis loop. Jia et al. (2024) introduces a language-based framework enabling effective zero-shot design in low-data regimes. Despite these advancements, these approaches essentially remain open-loop workflows, lacking the dynamic, knowledge-guided feedback required for more efficient and adaptive exploration.

### 2.2 LLM-BASED SCIENTIFIC AGENTS

Recent advancements in LLM have catalyzed sophisticated AI agents to accelerate scientific discovery(Bai et al., 2025; Wei et al., 2025; Hu et al., 2025). One strategy involves integrating expert tools: Bran et al. (2023); Kang & Kim (2024) enhance LLM performance in chemistry for tasks like synthesis and drug discovery, while Ruan et al. (2024) automates the entire synthesis workflow using six specialized GPT-4 agents. Another approach emulates the scientific process itself; Gottweis et al. (2025) system collaboratively generates, critiques, and refines hypotheses. Concurrently, other projects build foundational capabilities: Chai et al. (2025) creates an agent architecture for difficult general scientific benchmarks, and AlphaEvolve, an evolutionary agent, relies on an automated evaluator to guide LLM-driven code mutations. While these agents excel with fixed tools or general heuristics, they lack a symbolic framework for programmatic material design. MatEvolve addresses this gap by introducing MEL and MEB, enabling a efficient exploration of chemical space.

## 3 METHOD

### 3.1 OVERALL ARCHITECTURE

MatEvolve is a synergistic symbolic–LLM framework that reconceptualizes the process of materials design. It replaces the conventional enumeration-screening method with an intelligent insight-exploration-validation paradigm. As depicted in Fig. 2, MatEvolve consists of three integral components: Materials Edit Language (MEL), Materials Edit Base (MEB), and Materials Edit Engine (MEE).

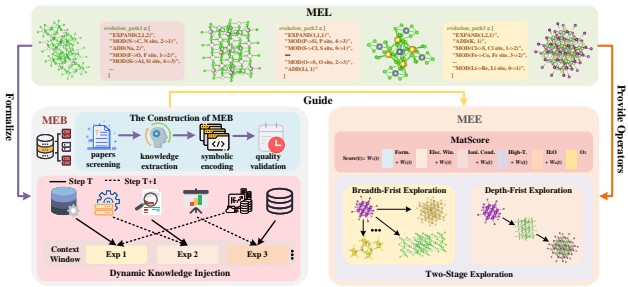

Figure 2: **Overall architecture**: MatEvolve framework operates on an iterative loop where the core MEE employs an AI agent to write MEL code, proposing precise material modifications. Guided by the MEB knowledge base, each new candidate is assessed by the MatScore fitness function, providing feedback to drive the next cycle of exploration.

**MEL:** MEL is a domain-specific symbolic language designed to formalize material design by encoding chemical operations (e.g. substitution, doping) into a programmable format. This formalism effectively codifies and makes explicit the heuristics prevalent in expert-guided materials design.

**MEB:** A structured knowledge base contains proven modification strategies from scientific literature, all represented in MEL. It provides the agent with validated insights to guide its exploration.

**MEE:** MEE is the core operational component that orchestrates the design process. It implements a closed-loop insight–exploration–validation to iteratively optimize material candidates. Guided

by MEB and chemical principles, the agent writes MEL code to propose syntactically valid material modifications. A two-stage exploration strategy is then employed, which transitions from a breadth-first exploration of diverse chemical spaces to an intensive, depth-first exploration within high-potential regions. Each resulting candidate is scored by MatScore, a dynamic fitness function that aggregates multi-objective metrics, including ionic conductivity, thermodynamic stability, and high-temperature stability. This score provides immediate feedback to guide MatEvolve's next exploration step, effectively closing the optimization loop.

## 3.2 MATERIALS EDIT LANGUAGE

Current atomic-level material doping modifications rely on operating directly on Crystallographic Information Files (CIFs), as modifications to chemical formulas cannot capture site-specific atomic occupancy. However, direct CIF manipulation is cumbersome and error-prone due to its data verbosity and complex symmetry constraints.

To address this challenge, as shown in Fig. 3, we introduce the **Composition Modification (MOD) operation** for the targeted, rule-based manipulation of CIFs. Its core functionalities include: 1) substitution of elemental species at designated crystallographic sites; 2) creation of vacancies to precisely control their concentration within the material; 3) automated adjustment of stoichiometry to maintain charge neutrality, tracking of all compositional changes.

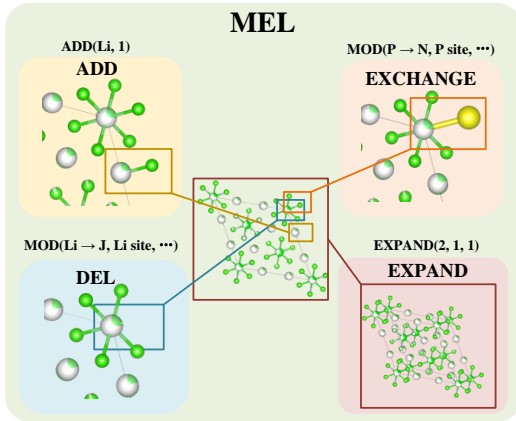

Figure 3: **MEL**: MEL achieves the precise modification of crystal structures using four fundamental operations: ADD, EXPAND, and the MOD operator, which performs both EXCHANGE and DEL.

The MOD operation circumvents the complexities of direct CIF file manipulation, enabling precise compositional control when transforming a pristine material into a target doped structure. We further leverage the fact that the Wyckoff sites provide an efficient method for characterizing atomic coordinates(Wyckoff, 1922; Goodall et al., 2022; Song et al., 2025). As all atoms on an equivalent Wyckoff site are related by symmetry operations, their positions can be derived from a single representative, significantly reducing data redundancy. Building on this principle, we propose the Material String representation, which condenses the crystal structure into a concise format: **Space group — Lattice — (Element – Wyckoff [Fractional Coordinate]) Sequence**. This representation preserves all core structural information, including space group symmetry, lattice parameters, and atomic site occupancies. It also provides an unambiguous textual description of the crystal structure, laying a robust foundation for synergy with the MOD operation and for efficient interpretation by machine learning models.

## 3.3 MATERIALS EDIT BASE

### 3.3.1 THE CONSTRUCTION OF MEB

LLMs possess strong language understanding and generation capabilities in general domains but face two core limitations in vertical materials domains like solid-state electrolytes: first, a scarcity of domain knowledge, as pre-training data struggles to cover fine-grained materials design rules (e.g., dopant element selection, ratio control); second, high randomness in the generation process, prone to producing operation schemes that violate chemical principles without domain knowledge constraints. To address these issues, this study constructs the Materials Edit Base (MEB), using the MEL symbolic system as a unified carrier to achieve structured integration and precise reuse of domain knowledge, providing targeted knowledge guidance for the MEE evolution engine.

The construction of MEB follows a closed-loop process of "data source screening → knowledge extraction → symbolic encoding → quality validation," ensuring the authority, accuracy and usability of the knowledge base. Focusing on the design of solid-state electrolyte materials, it systematically collects approximately 2000 academic papers published in the last decade in top tier journals such as

*Advanced Materials*, *Energy & Environmental Science*, and *Chemistry of Materials*. Next, a hybrid "automated initial screening + manual refinement" knowledge extraction scheme is designed to balance efficiency and precision, batch-extracting text snippets associating "operations–performance" from the papers. Finally, using the MEL symbolic system as a unified format, the extracted domain knowledge is converted into structured code parsable by LLMs.

Validation is performed on the symbolically encoded knowledge entries to ensure that the MEL code can be parsed by the decoder without grammatical errors. This results in approximately 200 high-quality domain knowledge entries, covering core design scenarios such as "dopant element selection," "ratio optimization," "vacancy control," and "lattice parameter adjustment."

### 3.3.2 DYNAMIC KNOWLEDGE INJECTION

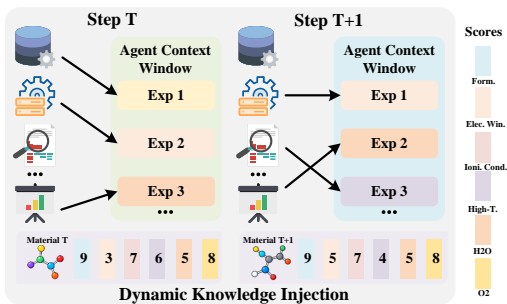

A static knowledge injection approach, utilizing a initial selected subset of MEB, was found to be suboptimal as it fails to adapt to the shifting performance bottlenecks of a material candidate throughout its exploration trajectory. As shown in Fig. 4, We implement a Dynamic Knowledge Injection mechanism that provides adaptive, context-aware guidance to the agent. This mechanism operates by first identifying the most deficient performance metric of the current leading candidate. Subsequently, it retrieves a curated, task-relevant subset of the top 30 knowledge entries from MEB to ensure the agent's efforts are precisely targeted at the most critical aspects of the design challenge, significantly accelerating convergence towards a multi-objective optimum.

Figure 4: **Dynamic knowledge injection**: An adaptive mechanism that accelerates material optimization by identifying the most critical performance bottleneck at each step and injecting targeted knowledge to guide the agent in resolving it.

### 3.4 MATERIALS EDIT ENGINE

#### 3.4.1 MATSCORE

To guide the design process, we developed MatScore, a unified multi-objective fitness function. MatScore provides rapid, quantitative feedback for the exploration by aggregating critical performance metrics.

Table 1: Material performance evaluation metrics reference.

| Metric | Meaning |
|---|---|
| $S_{val}$ | Proportion of valid CIF files |
| $S_{form}$ | Thermodynamic stability score based on Mattersim potential |
| $S_{elec}$ | Electrochemical window stability score |
| $S_{ion}$ | Li-ion transport capability score |
| $S_{highT}$ | Thermodynamic stability score at specified high temperatures |
| $S_{H_2O}$ | Resistance to decomposition in water environments |
| $S_{O_2}$ | Resistance to oxidation/decomposition in oxygen environments |
| $S_{SSE}$ | Composite score for electrolyte performance |
| $S_{elastic}$ | Material deformation resistance and mechanical strength score |
| $S_{barrier}$ | Ease of Li-ion diffusion score |
| $S_{stab}$ | Structural and performance stability during delithiation |
| $S_{gap}$ | Suitability of electronic bandgap for battery applications |
| $S_{Cathode}$ | Composite score for cathode performance |

Table 1 outlines key metrics to assess battery material performance. These metrics collectively evaluate structural, electrochemical, and mechanical properties of a material, guiding the selection and optimization of materials for enhanced battery performance and stability. All scores are z-normalized, sigmoid-mapped to [0,1], and integrated for a comprehensive material assessment.

$$S_{\text{SSE}} = \frac{1}{6} \sum_i \frac{1}{1 + e^{-S_i}} \quad \text{where } i \in \{\text{form, elec, ion, highT, } H_2O, O_2\} \tag{1}$$

$$S_{\text{Cathode}} = \frac{1}{4} \sum_i \frac{1}{1 + e^{-S_i}} \quad \text{where } i \in \{\text{elastic, stab, gap, barrier}\} \tag{2}$$

### 3.4.2 Evolution Strategy

To navigate the vast chemical space effectively, we designed a two-stage exploration strategy that explicitly balances global exploration with local exploitation. This approach addresses challenges where initial and broad modifications yield rapid diversification but often lead to premature convergence on performance plateaus. An overview is shown in Fig. 5.

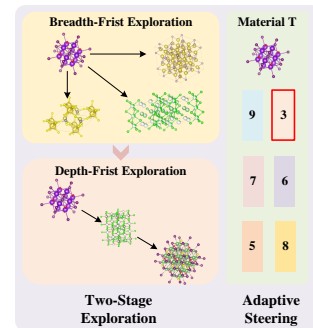

The initial Breadth-First exploration phase is designed to rapidly map the viable design space. This is achieved by employing high-variance, stochastic MEL operators, such as random elemental substitutions and broad-range adjustments to doping ratios. The objective is not to pinpoint a local optimum, but to efficiently discover diverse and structurally stable material families, thereby pruning vast, non-viable regions of the exploration space and establishing promising territories for further investigation. Upon identifying these promising subspaces, the strategy transitions to a Depth-First Exploration phase. The focus shifts to targeted refinement using fine-grained, deterministic MEL operators, like precise elemental tuning and incremental adjustments to composition. This approach minimizes redundant computation and accelerates convergence toward superior material configurations within the identified high-potential regions.

Figure 5: **Evolution strategy**: A two-stage exploration, balancing broad exploration with deep refinement, which is adaptively steered by the current material's performance bottleneck.

To steer this process, we implement an adaptive steering mechanism. At each exploration step, the system analyzes the normalized components of the MatScore to identify the primary performance bottleneck of the current leading candidate. This information dynamically directs the agent's generative focus, prompting it to prioritize operations that specifically address this weakness. This dynamic re-focusing ensures a balanced, multi-objective improvement path, creating a highly efficient and responsive closed-loop optimization process.

## 4 Experiments

### 4.1 Implementation details

Our implementation centers on LLM-generated "SEARCH/REPLACE diff" targeting the evolution path sequence, using custom MEL operators (EXPAND, MOD, ADD) under stoichiometric/Wyckoff constraints—unlike AlphaEvolve, our implementation integrates a symbolic system for the materials domain, extra knowledge injection, and dynamic exploration.

We uniformly use GPT-3.5 (Ouyang et al., 2022) Turbo as the main LLM for our experiments. During **Breadth-Frist Exploration**, temperature is set to 0.8 and top-$p$=0.9 to enhance operational diversity, supporting broad exploration; **Depth-Frist Exploration** reduces temperature and top-$p$ to 0.3 to improve generation determinism, facilitating precise convergence.

The evolutionary process adopts an island-based population structure with archive retention to regulate exploration. For **Breadth-First Exploration**, we set the population size to 200, archive size to 30, and number of islands to 6, with an elite ratio of 0.05 and an exploitation ratio of 0.3, enabling efficient exploration of large chemical spaces and parallel mining of diverse candidates. In contrast, **Depth-First Exploration** adjusts these parameters to a population size of 150, archive size of 20, and 3 islands, with an elite ratio of 0.1 and an exploitation ratio of 0.8, thereby focusing on fine-grained tuning in high-potential regions and achieving steady performance improvements.

We validate MatEvolve's effectiveness on two prominent, well-established tasks: solid-state electrolytes (SSE) using representative $Li_6PS_5Cl$ (focusing on MatScore core properties) and cathode materials using commercial precursors $LiCoO_2$ and $LiFePO_4$ (emphasizing energy-density-related properties and structural stability). Both tasks reuse the same evolution framework and scoring strategy for reproducibility and cross-system comparability.

## 4.2 COMPARATIVE EXPERIMENTS

### 4.2.1 COMPARISON WITH EXISTING METHODS

To assess the advantage of MatEvolve over hierarchical screening pipelines, we instantiate a baseline that combines local enumeration with hierarchical screening using the surrogate models in MatScore (full configuration in Appendix A) and we also compare against LLMatDesign (Jia et al., 2024). As shown in Table 2, MatEvolve improves the combined score by 15.6% over the screening baseline. Compared to LLMatDesign, MatEvolve improves the combined score by 19.9% and yields a 57.8% increase in the fraction of usable structures. These gains stem from MatEvolve's finer-grained symbolic interface (MEL) and its more flexible knowledge-injection (MEB) and exploration strategy (MEE).

Table 2: **Comparative experiment**: In the solid-state electrolyte (SSE) design task, $S_{val}$ and $S_{SSE}$ are the most critical metrics and also the key focus of this task. Results show that our proposed MatEvolve framework achieves the optimal performance, with the highest values in both $S_{val}$ and $S_{SSE}$, significantly outperforming the traditional Screening method and LLMatDesign.

| Config | Object | $S_{val}$ | $S_{SSE}$ | $S_{form}$ | $S_{elec}$ | $S_{ion}$ | $S_{highT}$ | $S_{H_2O}$ | $S_{O_2}$ |
|---|---|---|---|---|---|---|---|---|---|
| **Screening** | CIF | – | 0.464 | 11.094 | 0.224 | -4.662 | 10.701 | -0.316 | -1.611 |
| **LLMatDesign** | Formula | 29.5 | 0.421 | 10.136 | 0.043 | -4.479 | 10.951 | -0.141 | -1.577 |
| **None** | CIF | 38.2 | 0.469 | 9.919 | 0.089 | -4.004 | 9.934 | -0.042 | -1.582 |
| **+MEL** | MOD | 87.2 | 0.505 | 9.083 | 0.079 | -4.123 | 9.098 | -0.043 | -1.187 |
| **+MEL+MEE** | MOD | 86.1 | 0.547 | 9.079 | 0.415 | -4.676 | 9.094 | -0.111 | -0.748 |
| **MatEvolve** | MOD | **87.3** | **0.620** | 9.218 | 0.426 | -4.663 | 9.214 | -0.535 | -0.736 |

### 4.2.2 COMPARISON BETWEEN LLMS

To evaluate the performance of different LLMs on materials deisgn, we compare a range of closed-source models, e.g., GPT, Gemini (Team et al., 2023), Claude, and open-source models, e.g., DeepSeek (Liu et al., 2024), Qwen (Yang et al., 2025; Team, 2024), GLM (Zeng et al., 2025), on the SSE design task. As shown in Fig. 6 , larger models such as GPT-5, Grok-4, and Qwen-3-MAX achieve combined scores above 0.6, indicating relatively strong performance. By contrast, models such as Gemini2.5-Flash, Claude-3.7 and DeepSeek-V3.1 show relatively weaker results on this task, suggesting that domain-specific knowledge plays a more decisive role than general coding capability. Notably, within our MatEvolve framework, the smaller GPT-3.5 Turbo

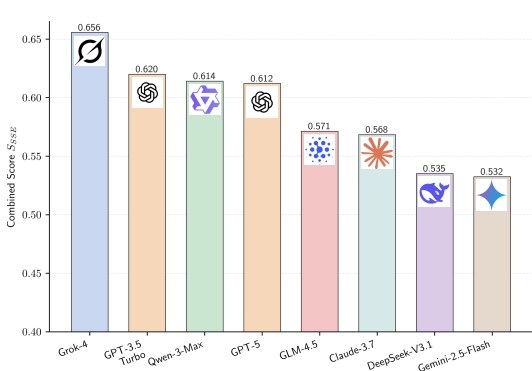

Figure 6: **$S_{SSE}$ comparison between LLMs**: Evaluate various closed-source and open-source LLMs on the SSE design task.Notably, the smaller GPT-3.5 Turbo performs strongly in our MatEvolve framework, indicating domain-specific knowledge is more critical than model scale.

(released in 2023) achieves unexpectedly strong performance, even outperforming GPT-5 in our evaluation. This observation indicates that, once a model has sufficient instruction-following capacity, specialized domain knowledge becomes more critical than scale-driven common-sense knowledge or programming ability for advancing materials design.

### 4.3 ABLATION STUDY

In this subsection, we ablate MatEvolve's three core components—MEL, MEB, and MEE—using a greedy protocol: (i) compare MEL against prior works and fix the best as the baseline; (ii) ablate MEB on this baseline and retain its best configuration; and (iii) ablate MEE on the resulting setup.

#### 4.3.1 ABLATION ON MEL

The symbolic system underpins programmatic material editing. To assess the effectiveness of **MEL**, we design two baselines: **Baseline 1**, adapted from LLMMatDesign, restricts edits to formula-level doping; **Baseline 2** directly prompts the LLM to modify CIF text. Fig. 7 reports average validity ($S_{\mathrm{val}}$) and overall performance ($S_{\mathrm{SSE}}$) under the setting without knowledge injection, two-stage exploration, or dynamic weighting.

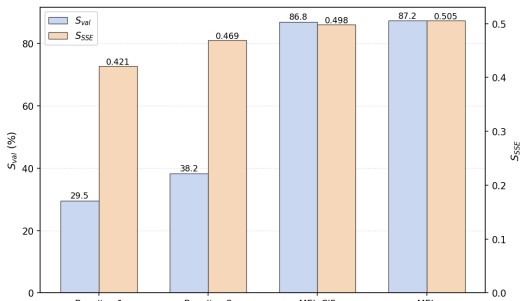

**MEL** substantially outperforms **Baseline 1** in both validity and combined score, showing that atom-level operations enable finer-grained and more reliable improvements. Compared with **Baseline 2**, **MEL** yields higher validity while efficiently representing edit operations, reducing context length and hallucination risk. In-

Figure 7: **Performance across symbolic systems**:By comparing the final results of our proposed MEL with the two baseline methods, MEL achieves superior performance, demonstrating that incorporating operators and Wyckoff positions can further enhance the effectiveness.

corporating Wyckoff positions further enhances both metrics, indicating that precise positional information is essential for guiding LLMs to design doping schemes aligned with chemical principles.

#### 4.3.2 ABLATION ON MEB

To assess the role of MEB, we compare knowledge injection strategies in Fig. 8. **None** uses MEL without knowledge; **MEB-static** adds 30 fixed prior entries; **MEB-dynamic** adaptively selects knowledge during optimization. Both injection strategies outperform the baseline, confirming the importance of expert knowledge. **MEB-dynamic** achieves the best final results, showing that on-demand knowledge selection is more effective than fixed injection.

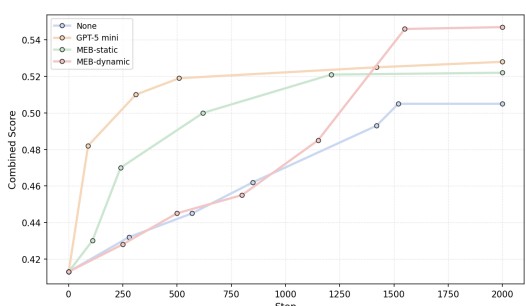

We further include GPT-5 mini under no injection as a stronger baseline. Compared with GPT-3.5 Turbo (**None**), **GPT-5 mini** improves more quickly in early stages and ultimately matches **MEB-static**, suggesting that larger LLMs act as implicit static knowledge injection via richer pretraining. Nonetheless, **MEB-**

Figure 8: **Performance comparison of different knowledge injection**: Compare no injection, MEB-static, MEB-dynamic, and GPT-5 mini (stronger LLM). Both MEB variants outperform no injection, and MEB-dynamic achieves the highest scores ( outperforming even GPT-5 mini) validating its context-aware advantage.

**dynamic**—with explicit, context-dependent knowledge injection—achieves the highest scores.

#### 4.3.3 ABLATION ON MEE

Fig. 9a shows the evolution of the combined score under different exploration strategies. **Single-Stage** applies only Breadth-First Exploration; **Two-Stage** switches to Depth-First Exploration after step 1500; and **Two-Stage+Weight** further introduces dynamic weighting. **Two-Stage** achieves higher final scores than **Single-Stage**, while **Two-Stage+Weight** accelerates progress and yields the best overall performance. Fig. 9b further illustrates how optimization focus shifts over time: early iterations emphasize performance-related metrics (ionic conductivity, electrochemical window), while later stages prioritize stability (water and high-temperature stability). This adaptive

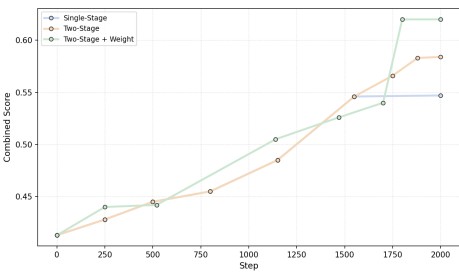 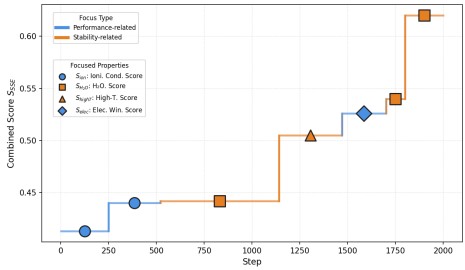

(a) $S_{SSE}$ of different evolution strategy.  (b) Focused metrics during dynamic evolution.

Figure 9: Ablation on MEE. Two-stage search with dynamic weighting yields the best overall performance, as MatEvolve adaptively balances performance and stability objectives across steps.

reweighting demonstrates the effectiveness of dynamic multi-objective optimization in guiding the exploration process.

### 4.4 EXTENDING MATEVOLVE TO CATHODE MATERIALS

Cathode materials critically determine battery energy density, cycle life, and safety. To assess the generalizability of MatEvolve beyond solid-state electrolytes, we extend the framework to cathode materials design, initializing from two widely deployed commercial systems: $LiFePO_4$ (LFP) and $LiCoO_2$ (LCO).

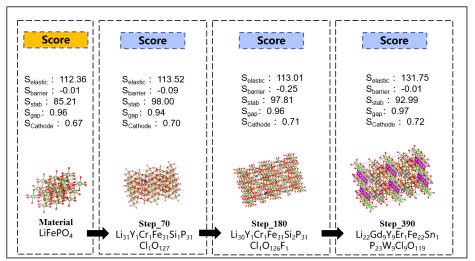 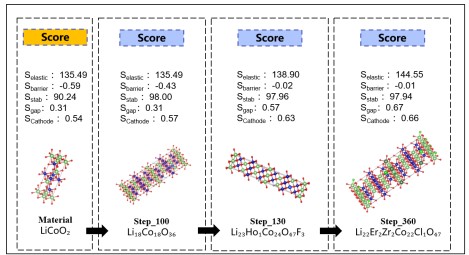

(a) Evolutionary trajectories for LFP-based cathodes.  (b) Evolutionary trajectories for LCO-based cathodes.

Figure 10: **MatEvolve on cathode materials.** Evolution pathways and performance profiles for LFP- and LCO-initialized candidates under MatEvolve; both systems exhibit consistent gains in the combined score, supporting the framework's generalizability in cathode design.

As shown in Fig. 10a, LFP-based candidates improve under multi-element co-doping (rare-earth, transition-metal, anionic): mechanical strength +17% and combined score from 0.67 to 0.72 (+0.05). For LCO (Fig. 10b), MatEvolve addresses high $Li^+$ diffusion barriers and a suboptimal band gap via Er/Zr dopants and Cl anions, raising the combined score from 0.54 to 0.67 (+0.12). See Appendix E for experimental setup and full numerical results.

## 5 CONCLUSION

In this work, we presented MatEvolve, a symbolic–LLM evolutionary agent that reframes materials design as a closed-loop insight–exploration–validation process. At its core, the Material Edit Language (MEL) enables atom-level symbolic operations, while the Material Editing Base (MEB) provides dynamic knowledge injection and the Material Evolution Engine (MEE) implements two-stage dynamic exploration. Applied to solid-state electrolytes and cathode materials, MatEvolve not only reproduces known pathways but also uncovers novel, chemically plausible candidates, achieving substantial gains over enumeration–screening methods. In the future, we will extend MatEvolve to parent-material discovery and synthesis-protocol design, and ultimately integrate these into a unified end-to-end materials scientific agent to accelerate the pipeline from candidate generation to practical preparation.

ETHICS STATEMENT

This work introduces MatEvolve, a symbolic–LLM evolutionary agent for materials design, aiming to accelerate the discovery of sustainable and high-performance materials. Our experiments rely on publicly available LLMs and datasets, without involving human subjects or private data, so privacy concerns are minimal. While the framework demonstrates significant scientific benefits, we acknowledge risks of potential misuse in designing harmful or hazardous compounds; to mitigate this, we emphasize that outputs must be experimentally validated and restrict release of hazardous generative capabilities. We commit to transparency and reproducibility through detailed documentation and code release (subject to safety constraints), and to fair attribution of prior work. Overall, MatEvolve highlights a paradigm shift in materials design while remaining mindful of safety, responsible deployment, and its broader societal impact.

REPRODUCIBILITY STATEMENT

To ensure reproducibility, our complete experimental configurations are provided in the supplementary material, including details of numeration-screening baseline (Appendix A), MEL details (Appendix B), MEB details (Appendix C), MEE details (Appendix D), details of cathode material design (Appendix E), and more visualization cases (Appendix F), are thoroughly documented.

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

# APPENDIX

## A    DETAILS OF NUMERATION-SCREENING BASELINE

We instantiate the traditional enumeration–screening baseline as a rational-design, funnel-style pipeline that first generates a very large pool of candidates and then filters them through staged thresholds. Starting from representative parent structures in the Li–P–S–Cl system, we construct approximately $4 \times 10^5$ candidates using special quasirandom structures (SQS): expert-curated whitelists bound the allowable dopants and their ranges on P and S sublattices; supercells are chosen to ensure integer occupancies, charge-balanced stoichiometry, and statistically disordered occupation on equivalent Wyckoff sites; vacancy cases are included with explicit compositional compensation. All structures are formula-normalized, de-duplicated, and validated for CIF parsability before screening. Screening proceeds hierarchically with the same surrogate suite as our main study to ensure comparability: (i) a fast sanity layer removes structures that fail parsing, violate elemental/dopant-set limits or per-site/total dopant caps, exhibit obvious valence inconsistencies, or break minimum interatomic-distance constraints; (ii) a primary-threshold layer evaluates the Mattersim-based formation-energy score $S_{\text{form}}$, electrochemical stability-window score $S_{\text{elec}}$ (phase-diagram chemical-potential domain), ionic-conductivity score $S_{\text{ion}}$ (multimodal DL predictor), high-temperature stability score $S_{\text{highT}}$ (free-energy correction via mixing entropy), and environmental stabilities $S_{\text{H}_2\text{O}}/S_{\text{O}_2}$ (lower bounds of competing-reaction energies). Each metric is $z$-normalized and mapped via a sigmoid to $[0, 1]$, and candidates must meet calibrated minimum per-metric thresholds (energy/processability first, then electrochemical and environmental robustness); (iii) a secondary ranking layer orders survivors by the task-specific composite objective (e.g., $S_{\text{SSE}}$ for electrolytes), with stricter single-metric tie-breakers (e.g., wider window or higher conductivity) resolving near ties. This open-loop, static, and non-adaptive workflow emphasizes breadth and simplicity—no dynamic knowledge injection, no feedback-driven edits, and no exploration–exploitation balancing—making it a strong large-scale baseline while remaining susceptible to threshold sensitivity and combinatorial sparsity in vast chemical spaces.

## B    DETAILS OF MATERIAL EDIT LANGUAGE (MEL)

To efficiently represent the doping process of materials, we propose the MEL notation system, which evolves the material from the initial structure to the target structure using a unified and executable symbolic framework. The material state is represented by a chemical formula with full stoichiometric expression: $F_0$ denotes the initial material, $F_i$ denotes the $i$-th intermediate, and $F_T$ denotes the final material (which may include doping elements), for example, $Li_7La_3Zr_2O_{12}$. The core evolution operator is MOD (composition modification): replacing the old element A with the new element B at the specified crystallographic site L, denoted as $\text{MOD}(A \rightarrow B, L, P_A \rightarrow Q_A, P_B \rightarrow Q_B)$, where $P_A/P_B$ and $Q_A/Q_B$ represent the stoichiometries before and after the operation, respectively, and $B = J$ indicates the creation of a vacancy. A single-step evolution is expressed as $F_i = F_{i-1} + \text{MOD}(\dots)$, and after serialization, it forms a complete evolution path, thereby connecting the continuous material space through discrete and auditable micro-operations. To ensure coordination and charge balance, the system introduces the $\text{EXPAND}(x, y, z)$ operation to control supercell expansion for stoichiometric scaling, and provides the $\text{ADD}(Li, num)$ operation for minimal charge compensation during non-isovalent substitutions (limited to Li addition and scaled consistently with the supercell factor). This notation system aligns strictly with stoichiometry, avoiding ambiguous placeholders; the evolution path combines human- and machine-readability, facilitating automatic verification and experimental reproducibility, for example

$$
\begin{aligned}
F_1 &= F_0 + \text{MOD}(\text{Zr} \rightarrow \text{W}, \text{Zr site}, 2.0 \rightarrow 1.75, 0 \rightarrow 0.25), \\
F_2 &= F_1 + \text{MOD}(\text{Li} \rightarrow \text{J}, \text{Li site}, 7.0 \rightarrow 6.5, 0 \rightarrow 0.5), \\
F_0 &= Li_7La_3Zr_2O_{12} \rightarrow F_2 = Li_7La_3Zr_{1.75}W_{0.25}O_{12}
\end{aligned}
\tag{3}
$$

At the same time, to describe the crystal structure in a concise textual representation, we design the Material String representation: $\text{SP} \mid a, b, c, \alpha, \beta, \gamma \mid (\text{AS} - \text{WS}[\text{WP}]) \rightarrow \cdots \rightarrow (\text{AS}_N - \text{WS}_N[\text{WP}_N])$. Here, SP is the space group number; $a, b, c, \alpha, \beta, \gamma$ are the lattice parameters; $(\text{AS} - \text{WS}[\text{WP}])$ represents an atom type: AS is the element symbol, WS is the Wyckoff site identifier (including multiplicity), and [WP] is the fractional coordinate. Multiple atoms are connected by "$\rightarrow$". This representation uses Wyckoff sites as the backbone, avoiding coordinate

redundancy caused by crystal symmetry; it significantly compresses the length while ensuring information completeness, supporting bidirectional reversible conversion between Material String and CIF/POSCAR, thereby enabling knowledge compression and efficient sequence modeling. For example, $Li_{24}P_4S_{20}Cl_4$ can be expressed as:

$$216 \mid 10.279, 10.279, 10.279, 90, 90, 90 \mid (Cl - 4a[0, 0, 0])$$
$$\to (P - 4b[0.5, 0, 0]) \to (S - 4c[0.25, 0.25, 0.25]) \tag{4}$$
$$\to (S - 16e[0.115, 0.384, 0.884]) \to (Li - 24g[0.25, 0.25, 0.023]).$$

This representation fully preserves key structural elements (group symmetry, lattice, and sites) in each evolution round, facilitating symbolic operations (such as site-specific replacements) and LLM's structural understanding of the operated materials.

In summary, the system starts by parsing the CIF/POSCAR file of $F_0$ into a Material String, and generates the evolution path (EXPAND/MOD/ADD) sequence in the EVOLVE-BLOCK module. Each step first applies the MOD operation at the stoichiometric level (including vacancy creation with $B = J$), uses EXPAND when necessary to ensure consistent supercell scaling, and utilizes ADD(Li, ·) for charge compensation; subsequently, the updated stoichiometry and site mappings are reflected back into the Material String, maintaining the symmetry framework and site semantics unchanged. Finally, the structure generator restores it to a CIF file: building the unit cell based on the lattice and space group, restoring atomic coordinates according to Wyckoff sites, and randomly placing newly added Li atoms into allowable sites with minimum distance constraints. After CIF output, systematic corrections are performed: ensuring uniqueness of _atom_site_label, removing redundant symmetry terms, revising metadata such as _chemical_formula_* and Z with parsed structure truths, and conducting secondary validation through a strict parser (requiring parsability, no fractional occupancies, and chemical formula consistency). The evaluator then performs multi-task scoring on the generated CIF and feeds back normalized indicators to the evolution loop, combined with rule library constraints (such as site restrictions, doping element whitelists and quantity limits, supercell scaling consistency) to achieve safe exploration. Thus, we construct a closed-loop paradigm of "symbolic operators (MOD/EXPAND/ADD) $\to$ Material String $\to$ CIF $\to$ evaluation feedback", unifying chemical interpretability, machine-readable verifiability, and structural reversibility in a lightweight and extensible methodological framework.

## C    DETAILS OF MATERIAL EDIT BASE (MEB)

### C.1    KNOWLEDGE EXTRACTION

This prompt extracts literature-grounded material evolution paths in a programmatic form, capturing the starting material, intermediate steps, and final products. For each step, it records the scientific purpose and a precise MOD/EXPAND/ADD operation string with site identifiers and stoichiometric changes, together with any quantitative impact when available. The output conforms to a strict, MEL-compatible JSON schema to ensure machine parsability, lossless provenance, and direct ingestion into the MEB for downstream dynamic knowledge injection and reproducibility.

```
## 1. Role and Goal
You are an expert AI assistant specializing in materials science
    and data extraction. Your task is to meticulously analyze the
    provided scientific text, identify descriptions of material
    synthesis or modification, and extract these "material
    evolution paths" directly into a structured JSON format.

## 2. Core Task
From the provided text, you will extract the starting material,
    the final material(s), and the sequence of modification
    steps, including the purpose and quantitative impact of each
    step.

## 3. Required JSON Output: Structure and Rules
**Your entire output MUST be a single JSON object.** Do not
    include any text, notes, or explanations outside of this
```

JSON. You must follow the structure and rules detailed below for each field.

## 4.Field-by-Field Rules:**

* `material_evolutions`: An array containing one or more evolution path objects. Create a new object for each independent path found in the text.
* `evolution_id`: A unique number for each evolution path, starting from 1.
* `starting_material`: An object describing the initial material.
    * `symbol`: Must be the string `"F_0"`.
    * `formula`: The chemical formula of the starting material, formatted with LaTeX-style subscripts (e.g., `H_{2}O`).
* `final_materials`: An array of objects for all final products of the evolution.
    * `symbol`: Must be the string `"F_T"`.
    * `formula`: The final chemical formula, using LaTeX-style subscripts.
* `evolution_path`: An array of objects, where each object represents a single, sequential step in the modification process.
    * `step_index`: The sequence number of the step, starting from 1.
    * `purpose`: A string describing the scientific reason for this step. Extract this from phrases like "in order to," "to achieve," etc.
    * `operation_string`: A string that precisely describes the modification. It **must** follow this format: `F_i = F_{i-1} + MOD(A->B, L, PA->QA, PB->QB)`.
        * `F_i`: The symbol for the material resulting from this step (e.g., `F_1`, `F_2`).
        * `F_{i-1}`: The symbol for the material from the previous step (e.g., `F_0`, `F_1`).
        * `MOD`: The modification operator.
        * `A->B`: The element `A` being replaced by element `B`. Use `J` for `B` if a vacancy is created.
        * `L`: The crystal site of the modification (e.g., `24d`, `96h`).
        * `PA->QA`: The change in stoichiometric coefficient for element `A`.
        * `PB->QB`: The change in stoichiometric coefficient for element `B`.
    * `quantitative_impact`: A string describing any measurable, numerical effect of the operation (e.g., `" = 1.2 mS/cm (from 0.8 mS/cm)"`). If no impact is mentioned, use the string `"No quantitative impact mentioned for this step"`.
    * `intermediate_material`: An object describing the material produced in this step.
        * `symbol`: The symbol for this intermediate (e.g., `F_1`). Must match the `F_i` in the `operation_string`.
        * `formula`: The full chemical formula of the intermediate material.
* `finalization_step`: A string that shows the final material `F_T` is equivalent to the result of the last step, e.g., `"F_T = F_2"`.

## 5. Task Execution

Now,please analyze the following content from the provided
    document.
Generate the JSON output strictly adhering to the structure and
    rules defined above.Please note that each operation in the
    material evolution path can only correspond to a replacement
    or a vacancy.If multiple elements are operated on,please
    separate them step by step.

## C.2   KNOWLEDGE INJECTION

Selects and ranks the Top-30 prior-knowledge entries from MEB that most directly improve the
current weakest objective in MatScore, ensuring chemical/site constraints and reproducibility.

```
# Core Task
- Given the normalized scores of the current best-performing
    material, identify the weakest metric among: Energy
    (stability), Electrochemical window length, Ionic
    conductivity (log, inverse-normalized), High-temperature
    stability, H2O resistance, and O2 resistance. Set this
    weakest metric as target_property. Propose evolution_path
    edits that primarily improve target_property while preserving
    the others, strictly obeying all chemical/site limits,
    supercell scaling (after EXPAND), and charge balance via
    ADD(Li).

# Knowledge Selection
- From the dopant-evolution knowledge base (e.g., merged
    literature evolutions / selected_knowledge.json), retrieve
    and rank items most relevant to target_property, then select
    the Top-30. Rank by:
-- Mechanistic alignment with target_property (e.g.,
    conductivity: S-site halide/oxygen substitution, mixed-anion
    disorder, Li-vacancy engineering; window: halide strategies
    that widen band gap without blocking Li pathways;
    energy/stability: near-isovalent P-site substitutions,
    reduced disorder; high_temp/H2O/O2: frameworks improving
    thermal/chemical robustness).
-- Element/site compatibility with constraints (P-site dopants
    from {Ti,Zr,Nb,Ta,Mo,W}; S-site from {O,Br,I,Cl}; Cl-site
    unchanged; total dopant types within limits).
-- Quantitative impact (magnitude of  in the target metric),
    reproducibility/clarity of operations (explicit
    MOD/ADD/EXPAND), and structural family proximity to LiPSCl
    systems (e.g., LGPS/argyrodite).
-- Practicality under Phase rules (Phase 1: diverse combos; Phase
    2: ratio-only fine-tuning with frozen dopant set and sites).
Output a concise Top-30 listeg:
- Evolution 1: Starting F_0 = Li_{6}PS_{5}Cl -> F_T =
    Li_{5.5}PS_{4.5}Cl_{1.5}; Path: MOD(S->Cl, 4c, 5.0->4.5,
    1.0->1.5) then MOD(Li->J, 48h, 6.0->5.5, 0->0.5); Purpose: To
    generate more Li+ vacancies and increase Cl-/S2- site
    disorder in order to increase ionic conductivity; Impact: No
    quantitative impact mentioned for this step.
- Evolution 2: Starting F_0 = Li_{10}GeP_{2}S_{12} -> F_T =
    Li_{10}SnP_{2}S_{12}; Path: MOD(Ge->Sn, Ge-site, 1.0->0.0,
    0->1.0); Purpose: to greatly reduce the raw material cost;
    Impact:  = 4 mS/cm.
- Evolution 3: Starting F_0 = Li_{6}PS_{5}Cl -> F_T =
    Li_{5.7}PS_{4.7}ClBr_{0.3}; Path: MOD(Cl->Br, 4d, 1.0->0.7,
```

```
0->0.3) then MOD(S->Cl, 4d, 5.0->4.7, 0.7->1.0) then
MOD(Li->J, 48h, 6.0->5.7, 0->0.3); Purpose: To induce
mixed-halide disorder for enhanced ion conduction and adjust
stoichiometry; Impact:  = 8.8 mS/cm (from 5.9 mS/cm).
```

# D   DETAILS OF MATERIAL EDIT ENGINE (MEE)

## D.1   DETAILED MATSCORE FOR SEE

### D.1.1   ENERGY STABILITY (GROUND STATE ENERGY AND FORMATION ENERGY)

Energy stability assessment is based on the total energy calculated using the Mattersim potential function and the derived formation energy to quantitatively characterize the thermodynamic stability of materials. The process is as follows: First, the CIF file is parsed into an ASE Atoms object, and the total energy $E$ of the system is directly calculated using the Mattersim potential function. The total number of atoms $N$ and the count of each element $n_i$ are then recorded. The reference chemical potentials $\mu_i$ are selected using a hierarchical strategy: for gaseous and non-metallic elements (such as O, N, H, etc.), the molecular reference states ($O_2$, $N_2$, etc.) are used as benchmarks, and $\mu_i$ is obtained by calculating the total energy of the corresponding molecule using the Mattersim potential function and dividing by the number of atoms; for other elements, the ASE's reference_states database is preferentially called, and the reference chemical potentials are also determined based on the energy calculation results of the standard state structures using the Mattersim potential function. The formation energy is calculated as the atomized form is The energy score is given by which is standardized and incorporated into the comprehensive assessment. A lower formation energy results in a higher score, providing a rigorous energetic criterion based on the Mattersim potential function for material screening.

$$
\begin{aligned}
E_{\text{form}} &= E - \sum_i n_i \mu_i, \\
E_{\text{form}}^{\text{atom}} &= \frac{E_{\text{form}}}{N}, \\
\text{energy\_score} &= -E_{\text{form}}^{\text{atom}}.
\end{aligned}
\tag{5}
$$

### D.1.2   ELECTROCHEMICAL STABILITY WINDOW (CHEMICAL POTENTIAL FEASIBILITY DOMAIN)

The electrochemical stability window employs a phase diagram-driven algorithm to solve the chemical potential feasibility domain, with the core being the screening of the chemical potential interval where the material is stable through thermodynamic reaction criteria. First, the reduced chemical formula (such as $A_x B_y C_z$) is automatically parsed and extracted from the CIF file of the target material. Subsequently, the Materials Project (MP) database cached data is called to aggregate competing phase data according to the chemical system of the material (such as Li-M-O), constructing a set containing all potential low-energy competing phases (such as $Li_2O$, $MO_2$, etc.). The calculation process takes the chemical potential of metallic Li, $\mu(\text{Li})$, as the core variable, traverses the value range of $\mu(\text{Li})$, and determines the interval endpoints $[\mu_{\text{high}}, \mu_{\text{low}}]$ where the material maintains thermodynamic stability. The window width $\Delta\mu$ is calculated as Its thermodynamic essence can be described through the reaction Gibbs free energy criterion: if for any combination of competing phases, the Gibbs free energy change ($\nu_i$ is the reaction stoichiometric coefficient, $\mu_i$ is the chemical potential of each phase) for the material decomposition reaction (such as $A_x B_y C_z \rightarrow aA + bB + cC$) satisfies $\Delta G > 0$, then the material will not decompose into lower-energy competing phases within this $\mu(\text{Li})$ interval, indicating electrochemical stability. The stability score is given by window_score $= \Delta\mu$; a larger window width indicates a broader applicable potential range and better performance. At the same time, to integrate into the multi-attribute comprehensive assessment framework, z-score standardization is used to eliminate dimensional effects, followed by mapping to the [0,1] interval via the Sigmoid function, enabling comparability and fusion with other performance indicators.

$$\Delta\mu = |\mu_{\text{high}} - \mu_{\text{low}}|,$$
$$\Delta G = \sum_i \nu_i \mu_i \tag{6}$$

### D.1.3 IONIC CONDUCTIVITY (MULTIMODAL DEEP LEARNING PREDICTION)

Ionic conductivity prediction employs the COmposition-Structure Bimodal Network (COSNet) multimodal deep learning model, using chemical composition and crystal structure as dual inputs for end-to-end prediction. The model first transforms the reduced chemical formula into a vector representation $C_i = g_C^r(c_i)$ through the composition branch (ROOST graph neural network), and the crystal structure parsed from the CIF into a vector representation $S_i = g_S^r(s_i)$ through the structure branch (de-CGCNN graph neural network). Then, attention mechanisms are used to compute weights ($w'_{ic}$, $w'_{is}$ after Softplus activation; when $s_i = s_{\text{null}}$, $w_{is} = 0$), obtaining a unified representation $M_i$ by element-wise summation or vector concatenation. Finally, an MLP outputs the logarithmic scale conductivity $\hat{y} = \log_{10} \sigma$ (unit S/cm), and the physical conductivity is restored via $\hat{\sigma} = 10^{\hat{y}}$. During training, data augmentation (supplementing composition samples without structure) promotes cross-modal representation alignment, combined with transfer learning (pre-training on a database of 18,000 Li-based compound bond valence barriers, then fine-tuning with 1,678 experimental conductivity data) and ensemble learning (model selection from 4 data subsets), addressing small-sample variance and extrapolation bias for new structures. In terms of performance, the test set MAE decreases from 1.022±0.047 in composition unimodal to 0.924±0.012, with prediction errors ¡1 order of magnitude for materials outside the training set. The score uses $\hat{y}$ (higher value indicates larger $\sigma$), incorporated into the comprehensive assessment via absolute value reversed z-score standardization, providing a basis for screening novel Li-ion conductors.

$$w_{ic} = \frac{w'_{ic}}{w'_{ic} + w'_{is}},$$
$$w_{is} = \frac{w'_{is}}{w'_{ic} + w'_{is}} \tag{7}$$

### D.1.4 HIGH-TEMPERATURE STABILITY (FREE ENERGY CORRECTION)

High-temperature stability assessment is based on the formation energy framework, introducing a mixing entropy term to construct a simplified free energy model to quantify temperature effects. The core formula is the atomized free energy where the mixing entropy $f$ is the effective mixing fraction, $x$ represents the component disorder, $k_B$ is the Boltzmann constant, and $T$ is the evaluation temperature. This model quickly captures the contribution of component disorder to stability at high temperatures through the $-T \cdot S_{\text{mix}}$ term, avoiding the high computational cost of full phonon spectrum calculations or heat capacity integrations. The interface returns high_temperature_stability = $G^{\text{atom}}(T)$, and during scoring, its negative ($-G^{\text{atom}}(T)$) is taken; a higher value indicates better thermodynamic stability at high temperatures. After standardization, it is incorporated into the multi-attribute comprehensive assessment system, prioritizing the screening of candidate structures suitable for high temperatures.

$$G^{\text{atom}}(T) = E_{\text{form}}^{\text{atom}} - T \cdot S_{\text{mix}},$$
$$S_{\text{mix}} = -f \cdot k_B \cdot [x \ln x + (1-x) \ln(1-x)] \tag{8}$$

### D.1.5 WATER STABILITY (COMPETING REACTION ENERGY WITH $H_2O$)

Water stability uses the lower bound of the thermodynamic driving force for the reaction between the material and $H_2O$ as a quantitative indicator, achieved through searching the most unfavorable reaction path. By constructing the "target material + $H_2O$" reaction system and combining the possible product set from the Materials Project cache (including hydroxides, oxides, etc.), the minimum per-atom reaction energy (unit eV/atom) is solved by traversing the reaction mixing ratio $r \in [0, 1]$. The reactant and product energies ($E_{\text{reactants}}$, $E_{\text{products}}$) are both calculated based on the Mattersim potential function, with reference chemical potentials selected consistently with the formation energy assessment. Physically, $\Delta E_{\text{min}}^{H_2O} > 0$ indicates thermodynamic stability of the material in

aqueous environments (no spontaneous decomposition tendency), while ¡0 indicates decomposition risk. The interface returns $\Delta E_{\text{min H}_2\text{O eV atom}}$, directly used as the scoring indicator (higher value indicates better stability), incorporated into the comprehensive assessment framework after z-score standardization and Sigmoid function mapping.

$$\Delta E_{\min}^{\text{H}_2\text{O}} = \min_r[E_{\text{products}}(r) - E_{\text{reactants}}(r)]/N \tag{9}$$

### D.1.6 OXYGEN STABILITY (COMPETING REACTION ENERGY WITH $O_2$)

Oxygen stability assessment adopts a thermodynamic framework consistent with water stability, merely replacing the environmental molecule with $O_2$. The core indicator is the lower bound of the most unfavorable reaction energy density (unit eV/atom), calculated by searching all possible reaction paths between the "material + $O_2$" system and oxygen-containing competing phases from the Materials Project cache (such as oxidation products, decomposition phases), based on the Mattersim potential function and $O_2$ molecular reference chemical potential ($\mu_O = E_{O_2}/2$). $\Delta E_{\min}^{O_2} > 0$ indicates thermodynamic tolerance of the material to oxygen environments (not prone to oxidation or decomposition), while ¡0 indicates spontaneous oxidation risk. The interface returns $\Delta E_{\text{min O}_2 \text{ eV atom}}$, used as the scoring indicator (higher value indicates better stability) and incorporated into the comprehensive assessment through z-score standardization and Sigmoid function processing, ensuring a unified and comparable multi-dimensional evaluation system with other performance indicators such as energy stability and electrochemical window.

$$\Delta E_{\min}^{O_2} = \min_r[E_{\text{products}}(r) - E_{\text{reactants}}(r)]/N \tag{10}$$

### D.2 EXPLORATION STRATEGY

### D.2.1 PROMPT CONFIGURATION FOR MATERIAL STRUCTURE OPTIMIZATION — PHASE 1

Drives breadth-first exploration to diversify element combinations, doping ratios, and cell sizes under strict system constraints, seeding promising regions for later refinement.

```
# Core Task & Role
You are a materials scientist specializing in solid-state
    electrolytes
and computational chemistry. Your task is to optimize material
    structures
by modifying the evolution_path to achieve higher scores.

# Material Edit Language & Initial Material Structure
- Supported operators:
  - EXPAND(x,y,z): Expand cell by x,y,z (x,y,z=1-3, Z20; start
      with EXPAND(1,1,1), adjust as needed).
  - MOD(A->B, L, PA->QA, PB->QB): Replace element A with B (B=J
      for vacancy) at site L.
    e.g., "MOD(P->Ti, P site, 4.0->3.6, 0->0.4)".
  - ADD(Li, num): Add num Li atoms for charge balance.

# Phase 1 Chemical System Constraints
- Allowed chemical systems: Li-P-S-Cl (base) or with doping
    elements:
  - P-site dopants: Max 2 from {Ti, Zr, Nb, Ta, Mo, W}.
  - S-site dopants: Max 2 from {O, Br, I, Cl}.
  - Total dopants: Max 4 (2 P-site + 2 S-site).
- Charge balance: Use ADD(Li, num) for non-isovalent
    substitutions, scaled with supercell expansion.
- Violation penalty: Systems exceeding limits are rejected.

# Phase 1 Search Strategy
- Perform breadth-first exploration to discover high-performing
    element
```

```
1080  combinations and ratios.
1081  - Combined doping: Explore P+S combinations, ensuring diversity.
1082  - Expansion: Use varied EXPAND ratios (x,y,z=1-3, Z20).
1083  - Doping count check: Before generating, verify P-site dopants 2,
1084    S-site
1085  dopants 2, total 4.
1086  - Bold exploration: Avoid repetitive patterns, try new ratios and
1087    elements
1088  to escape local optima.
1089
1090  # Output Requirement
1090  - Generate only SEARCH/REPLACE diffs for the evolution_path in
1091  initial_program.py (between # EVOLVE-BLOCK-START and #
1092    EVOLVE-BLOCK-END).
1093  - Format:
1094  <<<<<<< SEARCH
1095  evolution_path = ["EXPAND(1,1,1)", "MOD(P->Ti, P site, 4.0->3.0,
1096    0->1.0)",
1097  "MOD(S->O, S site, 20.0->19.0, 0->1.0)"]
1098  =======
1099  evolution_path = ["EXPAND(x,y,z)", "MOD(P->Element1, P site,
1100    q1->q2, 0->a1)", ...]
1101  >>>>>>> REPLACE
1102
1103
1104  D.2.2  PROMPT CONFIGURATION FOR MATERIAL STRUCTURE OPTIMIZATION — PHASE 2
1105
1106  Performs local refinement around the best Phase 1 program by mildly adjusting supercell sizes and
1107  dopant ratios while freezing dopant sets and sites.
1108
1109  # Core Task & Role
1110  You are a materials scientist specializing in solid-state
1111    electrolytes and computational chemistry. Your task is to
1112    refine material structures from Phase 1 by modifying the
1113    evolution_path to achieve higher scores.
1114
1115  # Material Edit Language
1116  - Supported operators:
1117   - EXPAND(x,y,z): Expand cell by x,y,z (x,y,z=1-2, Z12; prefer
1118     mild expansions from Phase 1).
1119   - MOD(A->B, L, PA->QA, PB->QB): Replace element A with B (B=J
1120     for vacancy) at site L, e.g., "MOD(P->Ti, P site, 4.0->3.6,
1121     0->0.4)".
1122   - ADD(Li, num): Add num Li atoms for charge balance (random
1123     positions, min_dist=1.5 ).
1124  - Use the dopant element set and doped sites from the selected
1125    Phase 1 program; do not introduce new elements or sites.
1126
1127  # Phase 2 Expansion and Ratio Adjustment
1128  - Freeze dopant elements and sites from the Phase 1 program.
1129  - Adjust doping ratios by 0.5 to 1.0 per element to explore local
1130    optima try.
1131  - Apply mild EXPAND changes (x,y,z=1-2, Z12) to adjust supercell
1132    size, scaling atom counts proportionally.
1133  - Charge balance: Use ADD(Li, num) for non-isovalent
        substitutions, scaling num with supercell expansion.
      - Verify: Resulting composition must match allowed systems.

      # Phase 2 Search Strategy
```

```
- Conduct deep, localized search around the selected Phase 1
    program to optimize doping ratios and supercell expansions.
- Ratio fine-tuning: Adjust doping quantities incrementally (0.5
    to 1.0) for P-site and S-site dopants.
- Expansion tuning: Test mild EXPAND variations (x,y,z=1-2, Z12),
    prioritizing small changes from Phase 1s expansion.
- Exploration: Focus on small, incremental changes to avoid
    drastic deviations; prioritize high-scoring configurations.

# Output Requirement
- Generate only SEARCH/REPLACE diffs for evolution_path in
    initial_program.py (between # EVOLVE-BLOCK-START and #
    EVOLVE-BLOCK-END).
- Format:
<<<<<<< SEARCH
evolution_path = ["EXPAND(1,1,1)", "MOD(P->Ti, P site, 4.0->3.0,
    0->1.0)", "MOD(S->O, S site, 20.0->19.0, 0->1.0)"]
=======
evolution_path = ["EXPAND(1,1,2)", "MOD(P->Ti, P site, 4.0->3.2,
    0->0.8)", "MOD(S->O, S site, 20.0->19.2, 0->0.8)", "ADD(Li,
    0.4)"]
>>>>>>> REPLACE
```

### D.2.3 WEAKEST-PROPERTY FOCUSING PROMPT

Sets optimization weights to prioritize the weakest normalized metric while preserving non-zero emphasis on all objectives; also instructs edits that target the identified bottleneck without violating constraints.

```
# Core Task
- Given the normalized scores of the material with the optimal
    current performance (higher scores indicate better
    performance), please make a decision based on the optimal
    scores.

# Decision Requirements
- Identify the weakest indicator = the indicator corresponding to
    the minimum value among the above six normalized scores
    (referred to as target_property).
- Set weights to prioritize the weakest item while keeping the
    weights of other items non-zero:(In case of tied scores,
    select the first item according to the following priority
    order: electrical conductivity -> window width -> energy ->
    high-temperature stability -> water resistance -> oxidation
    resistance.)

# Operational Guidelines:
- Propose a modified scheme for the evolution path, focusing on
    primarily improving the target property while avoiding
    significant deterioration of other properties.
- Must always comply with chemical/site constraints, the scaling
    rule after EXPAND, and the charge balance requirement
    achieved by ADD(Li) (lithium addition).
```

# E DETAILS OF CATHODE MATERIAL DESIGN

## E.1 DETAILED MATSCORE FOR CATHODE

### E.1.1 ELASTICITY SCORE (SHEAR MODULUS, G_VRH)

We evaluate the material's resistance to shear deformation by estimating the polycrystalline shear modulus using a universal machine-learned interatomic potential (MLIP). Starting from a CIF structure, we generate small, symmetry-preserving strains and obtain the corresponding stress responses to assemble the elastic stiffness tensor $C_{ij}$ (Voigt notation, $6 \times 6$). From $C_{ij}$, we compute the Voigt and Reuss bounds for the shear modulus of an effective polycrystal, which capture the upper and lower limits under uniform strain and uniform stress assumptions, respectively. The Voigt–Reuss–Hill (VRH) average $G_{\mathrm{VRH}}$ then provides a widely accepted effective shear modulus for isotropic polycrystals. MLIP backends typically report elastic quantities in energy-density units (eV/Å$^3$) owing to their atomistic nature; we therefore convert to SI-consistent GPa. In practice, larger $G_{\mathrm{VRH}}$ indicates stronger resistance to shear, enhanced rigidity, and improved mechanical robustness, which is desired for structural integrity under cycling and processing. When structures are partially disordered, the elastic inversion can be less reliable; we mitigate this by (i) small-strain linear regime, (ii) consistent strain grid, and (iii) unit conversion with a fixed factor.

$$
\begin{aligned}
G_V &= \frac{1}{15}\Big(C_{11} + C_{22} + C_{33} - C_{12} - C_{13} - C_{23} + 3(C_{44} + C_{55} + C_{66})\Big), \\
G_R &= \frac{15}{4\big(S_{11} + S_{22} + S_{33} - S_{12} - S_{13} - S_{23}\big) + 3\big(S_{44} + S_{55} + S_{66}\big)}, \\
G_{\mathrm{VRH}} &= \frac{G_V + G_R}{2}.
\end{aligned}
\tag{11}
$$

### E.1.2 BARRIER SCORE (LI-ION DIFFUSION BARRIER VIA NEB)

We quantify Li-ion mobility through the activation barrier computed by the climbing-image nudged elastic band (CI-NEB) method. The initial and final states are built by a Li-vacancy hop between the nearest pair of Li sites automatically detected in the structure. Intermediate images are placed by IDPP interpolation to yield a smooth initial path. All images share the same MLIP force field and are relaxed with BFGS under a force threshold, while the highest-energy image climbs to the saddle point. The barrier is the energy difference between the maximum along the path and the lower of the two endpoints, consistent with transition-state theory. For scoring, we apply a monotonic transformation that rewards lower barriers (faster diffusion) by taking the negative value, resulting in higher scores for smaller $E_{\mathrm{barrier}}$. This definition is simple, scale-aware, and preserves the relative ranking across candidates.

$$
\begin{aligned}
E_{\mathrm{barrier}} &= \max_i E_i - \min\big(E_{\mathrm{initial}}, E_{\mathrm{final}}\big), \\
\mathrm{Barrier\_Score} &= -E_{\mathrm{barrier}} \ (\mathrm{eV})
\end{aligned}
\tag{12}
$$

### E.1.3 STABILITY SCORE (DELITHIATION THERMODYNAMICS AND VOLTAGE)

We probe thermodynamic stability against delithiation by sampling configurations with $k$ Li removed from the host lattice (systematic selection for determinism), relaxing each structure, and recording the total energy $E(x)$ at Li fraction $x$ (with $n(x)$ Li per simulation cell). Using the fully delithiated host energy $E(\mathrm{host})$ and the Li chemical potential $\mu_{\mathrm{Li}}$ obtained from bulk bcc Li (per-atom energy), we define a formation-energy-like quantity $\Delta E(x)$ relative to (host + Li metal). Constructing the lower convex hull of $\Delta E(x)$ identifies the stable compositions (hull vertices); the vertical distances from the hull, $d_{\mathrm{hull}}(x)$, measure metastability. Between adjacent stable compositions $x_1 < x_2$, the two-phase average voltage follows from the reaction free-energy slope (with internal energies used as a proxy at $T = 0$ K). We then aggregate a 0–100 stability score from four components: (i) coverage of the stable $x$-range (40%), (ii) thermodynamic stability via a sigmoid of the average $|d_{\mathrm{hull}}|$ (30%), (iii) voltage quality and smoothness favoring practical ranges and low variance (20%), and (iv) a phase-count prior that prefers a small number of well-defined phases

(10%). This composite captures the breadth of stable compositions, their depth below the hull, electrochemical viability, and parsimony of phase evolution.

$$\Delta E(x) = E(x) - E(\text{host}) - n(x)\,\mu_{\text{Li}},$$
$$d_{\text{hull}}(x) = \Delta E(x) - \Delta E_{\text{hull}}(x),$$
$$V(x_1 \to x_2) = -\frac{E(x_2) - E(x_1) - \big(n(x_2) - n(x_1)\big)\mu_{\text{Li}}}{n(x_2) - n(x_1)}.$$

(13)

### E.1.4  BANDGAP SCORE (TARGETED ELECTRONIC SUITABILITY)

We map the band gap $E_g$ to a normalized suitability score using a Gaussian kernel centered at 2.0 eV with standard deviation 1.0 eV. This choice reflects the qualitative design target for battery electrode materials: very small gaps risk electronic shorting or parasitic conduction (metallic behavior), while very large gaps may impede electronic transport and limit rate capability; an intermediate gap near $\sim$2 eV is often desirable in practice, especially in conjunction with conductive additives. The Gaussian mapping is smooth, bounded in $[0, 1]$, and provides a robust, differentiable measure that penalizes large deviations from the target without hard thresholds.

### E.2  EXPERIMENT SETUP

We conduct diff-based evolutionary structure editing on LiFePO$_4$ (olivine) under strict chemical-system constraints to maximize a composite objective (higher Elasticity, Barrier-score via lower diffusion barrier, Stability, and Bandgap suitability) while ensuring diversity and reproducibility. The search uses a single phase with up to 200 iterations and checkpoints every 10 iterations; an initial population of 300, 8 parallel islands, and an archive of 50 to preserve diverse high-quality candidates; an elite retention ratio of 0.03 and exploitation ratio of 0.20 guide selection pressure; two concurrent evaluations (timeout 300 s) balance throughput and stability; logging at INFO ensures traceability. Generation randomness is steered by a temperature of 1.2 together with a simulated-annealing acceptance schedule: the edit-acceptance temperature starts at 1.0 and decays by $\times 0.9$ every 10 iterations to $\sim 0.2$, encouraging broad exploration early and convergence later; if no frontier improvement is observed for 20 iterations, we trigger a controlled restart by resampling 30% of the population. Cell size is controlled by `EXPAND(x,y,z)` with $x, y, z \in \{1, 2, 3\}$ and total supercell size $Z \leq 20$. Site-specific dopant limits are strictly enforced (each of Li/Fe/P/O sites $\leq 2$ unique dopant elements; TOTAL unique dopants $\leq 6$), and any non-isovalent substitution must be charge-balanced using `ADD(Li, num)` scaled proportionally with the supercell; ordered occupancies (occ=1) are mandatory to avoid parsing issues. A two-stage cascade screening (thresholds $[0.50, 0.75]$) improves robustness and throughput. Early iterations prioritize chemical diversity (broader EXPAND usage and element rotation), while later iterations emphasize local refinement (higher MOD frequency, reduced EXPAND). Periodic inter-island migration (every 20 iterations, $\sim 5\%$ of individuals) mitigates premature convergence and disseminates promising strategies, and the archive jointly optimizes score-frontier quality and chemical-space representativeness under the above constraints.

### E.3  PROMPT

Before running cathode experiments, we use the following task prompt to steer LiFePO$_4$ (LFP) optimization under strict dopant/site limits. It emphasizes mechanical robustness, diffusion kinetics (lower barrier), delithiation stability, and electronic suitability.

```
# Core Task & Role
You are a materials scientist specializing in battery cathode
    materials and computational chemistry.
Your task is to optimize LiFePO4-based structures by modifying
    the evolution_path to achieve higher
scores: Elasticity_Score (), Barrier_Score ( by lower barrier),
    Stability_Score (), and Bandgap_Score ().

# Initial Material & Edit Language
```

```
1296    - Initial structure: LiFePO4 (olivine, 4 Li, 4 Fe, 4 P, 16 O per
1297        unit cell), see initial_program.cif.
1298    - Supported operators:
1299     - EXPAND(x,y,z): Expand the cell by x,y,z, with x,y,z  {1,2,3}
1300        and total Z  20.
1301     - MOD(A->B, L, PA->QA, PB->QB): Replace element A with B (B=J
1302        for vacancy) at site L.
1303      Example: "MOD(Fe->Na, Fe site, 4.0->3.5, 0->0.5)".
1304     - ADD(Li, num): Add Li for charge balance. num MUST scale with
1305        supercell expansion.
1306
1307    # Strict Chemical System Constraints (LiFePO4 base)
1308    - Only the following site-specific dopants are allowed (each site
1309        2 unique elements; TOTAL  6):
1310     - Li-site dopants (2): {Si, Er, Ho, Yb, Gd}
1311     - Fe-site dopants (2): {W, Sn, Cr} (n-type) OR {Na, K, Ag, Cu}
1312        (p-type)
1313     - P-site dopants (2): {Si, Ge, Y, B}
1314     - O-site dopants (2): {F, Cl} (n-type) OR {N} (p-type)
1315    - TOTAL unique dopants across all sites must be  6 at all times.
1316    - Charge balance is crucial: for any non-isovalent substitution,
1317        you MUST use ADD(Li, num),
1318      and num MUST scale with EXPAND(x,y,z).
1319    - Maintain ordered structures (occ=1). Any violation leads to
1320        immediate rejection and resampling.
1321
1322    # Search Strategy (Phase 1)
1323    - Encourage diversity early: vary EXPAND ratios (still Z  20),
1324        rotate through allowed dopants per site,
1325      and explore different stoichiometric ratios. Avoid repetitive
1326        patterns.
1327    - Combined doping: Explore multi-site co-doping (e.g., Li+Fe,
1328        Fe+P, P+O), but NEVER exceed per-site and TOTAL limits.
1329    - Doping count check BEFORE emitting any diff:
1330     - Count Li-site dopants:  2
1331     - Count Fe-site dopants:  2
1332     - Count P-site dopants:  2
1333     - Count O-site dopants:  2
1334     - TOTAL dopants:  6
1335     - If any limit is exceeded, reduce dopants starting with the
1336        least promising combinations.
1337    - Charge balance enforcement: whenever non-isovalent MOD is used,
1338        immediately add ADD(Li, num) scaled by EXPAND.
1339    - Use bold changes to escape local optima, but keep the system
1340        valid.
1341
1342    # Output Requirement (STRICT)
1343    - Only output SEARCH/REPLACE diffs that modify the evolution_path
1344        list inside initial_program.py
1345      (between the exact markers "# EVOLVE-BLOCK-START" and "#
1346        EVOLVE-BLOCK-END").
1347    - Do NOT output any explanations or comments. Maintain EXACT
1348        indentation and formatting.
1349    - Format:
        <<<<<<< SEARCH
            evolution_path = [
                "EXPAND(1,1,1)",
            ]
        =======
```

```
evolution_path = [
    "EXPAND(x,y,z)",
    "MOD(Li->Element1, Li site, q1->q2, 0->a1)",
    "MOD(Fe->Element2, Fe site, q3->q4, 0->a2)",
    "MOD(P->Element3, P site, q5->q6, 0->a3)",
    "MOD(O->Element4, O site, q7->q8, 0->a4)",
    "ADD(Li, num_scaled_by_expand)",
]
>>>>>>> REPLACE
```

## E.4 DETAILED RESULTS

Table 3: **Evolution of Li-based sulfide electrolyte materials**: The table shows the performance variation of sulfide electrolytes during the evolutionary optimization process, with $S_{\text{SSE}}$ (composite electrolyte performance score) as the core evaluation metric.

| Evolution Step | Chemical Formula | $S_{\text{SSE}}$ | $S_{\text{form}}$ | $S_{\text{elec}}$ | $S_{\text{ion}}$ | $S_{\text{highT}}$ | $S_{\text{H}_2\text{O}}$ | $S_{\text{O}_2}$ |
|---|---|---|---|---|---|---|---|---|
| Step 1 | Li12P4S16 | 0.41 | 11.74 | 0.47 | -6.04 | 11.46 | -0.56 | -1.06 |
| Step 2 | Li11P3Si1S16 | 0.48 | 11.02 | 0.18 | -5.07 | 10.79 | -0.32 | -0.86 |
| Step 3 | Li75As2P22S93O3 | 0.51 | 11.10 | 0.27 | -4.94 | 11.01 | -0.33 | -0.79 |
| Step 4 | Li25As1P7S32 | **0.57** | 10.48 | 0.37 | -4.44 | 10.29 | -0.17 | -0.69 |

Table 4: **Evolution of Li-based cathode materials**: The table presents the performance evolution of Li-based cathode materials, with $S_{\text{elastic}}$ (mechanical strength), $S_{\text{stab}}$ (delithiation stability) and $S_{\text{Cathode}}$ (composite cathode performance) as key metrics.

| Evolution Step | Chemical Formula | $S_{\text{Cathode}}$ | $S_{\text{elastic}}$ | $S_{\text{stab}}$ | $S_{\text{barrier}}$ | $S_{\text{gap}}$ |
|---|---|---|---|---|---|---|
| Initial | LiFePO4 | 0.67 | 112.36 | 85.21 | -0.01 | 0.96 |
| Step 70 | Li31Y1Cr1Fe31Si1P31Cl1O127 | 0.70 | 113.52 | 98.00 | -0.09 | 0.94 |
| Step 180 | Li30Y1Cr1Fe31Si2P31Cl1O126F1 | 0.71 | 113.01 | 97.81 | -0.25 | 0.96 |
| Step 390 | Li22Gd9Y9Er1Fe22Sn1P23W9Cl9O119 | **0.72** | **131.75** | 92.99 | -0.01 | 0.97 |

Table 5: **Evolution of LiCoO-based materials**: The table displays the performance optimization process of LiCoO-based cathode materials, focusing on the improvement of $S_{\text{gap}}$ (bandgap suitability) and $S_{\text{Cathode}}$ (composite cathode performance).

| Evolution Step | Chemical Formula | $S_{\text{Cathode}}$ | $S_{\text{elastic}}$ | $S_{\text{stab}}$ | $S_{\text{barrier}}$ | $S_{\text{gap}}$ |
|---|---|---|---|---|---|---|
| Initial | LiCoO2 | 0.54 | 135.49 | 90.24 | -0.59 | 0.31 |
| Step 100 | Li18Co18O36 | 0.57 | 135.49 | 98.00 | -0.43 | 0.31 |
| Step 130 | Li23Ho1Co24O47F3 | 0.63 | 138.90 | 97.96 | -0.02 | 0.57 |
| Step 360 | Li22Er2Zr2Co22Cl1O47 | **0.66** | **144.55** | 97.94 | -0.01 | **0.67** |

## F VISUALIZATION CASES

As shown in Fig. 11, MatEvolve, when grounded in literature-derived domain knowledge, successfully reproduces and further extends multiple reported effective doping pathways. The selected dopant species and site-occupancy strategies are consistent with conclusions from the original studies, supporting the chemical plausibility and consistency of the learned edits. A representative example demonstrates our successful reproduction of a reported pathway ((Zhou et al., 2019)) at step 60, including the executed MEL operations and the resulting property trends, thereby evidencing the framework's effectiveness and practical viability in knowledge-grounded materials design.

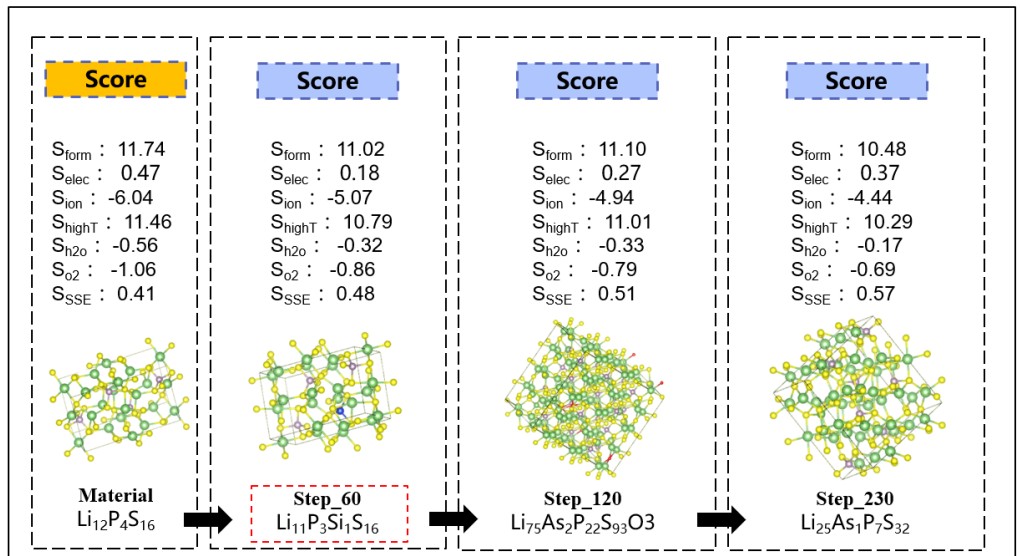

Figure 11: **Reproduced doping pathways**. MatEvolve successfully reproduces the correct literature-reported doping/evolution paths and their expected property trends, providing direct evidence that the framework is effective and practically viable.

## G  THE USAGE OF LLMS

Large Language Models (LLMs) were employed solely as assistive tools during the preparation of this manuscript. Specifically, LLMs were used to improve grammar and clarity, help summarize related literature, and refine the expression of concepts in figures. All core research ideas, experimental design, analyses, and conclusions were developed entirely by the human authors, who take full responsibility for the originality, validity, and final content of this paper.

