# OpenReview forum: "MatEvolve: A Synergistic Symbolic–LLM Agent for Multi-Objective Materials Design"
_ICLR.cc/2026/Conference — ICLR 2026 Conference Withdrawn Submission_

### Official Review · Reviewer_sjSF · 2025-10-20

**Soundness:** 3
**Presentation:** 3
**Contribution:** 3
**Rating:** 6
**Confidence:** 3

**Summary:**

The paper proposes synergistic symbolic LLM framework MatEvolve for material design. MatEvolve uses Material Editing Language (MEL) to edit design. It incorporates Material Editing Base (MEB) and Material Evolution Engine (MEE) as two core components.

**Strengths:**

1. The design of the components MEL, MEB and MEE are novel. The author also proves the effectiveness of these modules through ablations.
2. Experiments on solid-state electrolyte shows promising results.
3. Sufficient ablations and qualitative results are provided for better understanding.

**Weaknesses:**

1. The proposed method relies on the constructed MEB to provide domain specific knowledge. It is unclear how the quality of MEB affects the overall performance of the framework.
2. The scope of the experiments is limited as the methods are only tested on solid-state electrolyte and cathod material design. I would appreciate if the author can report more results on other material design tasks.
3. The experiments only compare MatEvolve against LLMatDesign. More baselines should be included for comprehensive evalutaion.

**Questions:**

Please address the concern as mentioned in weakness part.

---

### Official Review · Reviewer_NVU7 · 2025-10-30

**Soundness:** 3
**Presentation:** 2
**Contribution:** 3
**Rating:** 4
**Confidence:** 4

**Summary:**

This paper proposes an agentic approach to materials design, addressing the inefficiency and computational prohibitiveness of traditional enumeration--screening approaches. The proposed approach resembles AlphaEvolve by taking an evolutionary strategy, where the design is mutated iteratively by chasing one or a few objectives. Novel to the approach is a symbolic representation of the mutation, which is described by a domain-specific language invented for material editing. The authors demonstrate a 32.2% improvement over direct material structure modification and a 33.6% improvement over the traditional enumeration--screening approach.

**Strengths:**

The novelty and main contribution of the work are the integration of a symbolic system into an AlphaEvolve-style of agentic workflow. Specifically, the authors design a formal language that describes the editing steps, which can improve over LLM-proposed edits based on only natural language. Additionally, the authors curate a knowledge base, in the format of the language, of exemplified edits from the literature. These exemplified edits serve as useful inspiration to guide the design.

The authors conduct a reasonably comprehensive set of experiments to evaluate the proposed method and compare it with relevant LLM-driven design methods. The ablation study carefully discusses the impacts of the three components of the method: the language, the knowledge base, and the evolution strategy. The authors also demonstrate two practical use cases: designing solid-state electrolytes and cathode materials.

**Weaknesses:**

Despite the novelty and good coverage, this paper's contribution is not convincingly strong because of the heavy influence of AlphaEvolve and the unpolished presentation. Specific comments are below.

The figures that illustrate the proposed method are unintuitive and hard to understand. The overall architecture (Figure 2) is not a workflow but an ensemble of concepts and components. The MEL illustration (Figure 3) is ok. The knowledge injection figure (Figure 4) is very cryptic; the reader does not have a clue what the boxes (such as "Exp 1") are and what the number sequences (such as 9, 3, 7, 6, 5, 8) mean. The evolution strategy figure (Figure 5) is also cryptic, especially the "adaptive steering" part. Overall, these figures do not help the reader's understanding of the work.

The section regarding knowledge base construction (Section 3.3.1) can benefit from more elaboration, particularly regarding what efforts are automatic (through LLM prompting) and what are manual. Also, the validation procedure should be given more details, such as what the "decoder" is and whether human experts are involved. After all, manually inspecting 200 entries sounds manageable and it is expected that the authors conduct expert validation.

It would be helpful to discuss the coverage (in terms of the kind of materials) of the 200 entries and their intersection with the two use cases in the paper.

The presentation of the evolution strategy (Section 3.4.2) is insufficient. The authors would want to discuss (1) what the differentiation is from AlphaEvolve; (2) why not use traditional optimization methods (such as genetic algorithms, Bayesian optimization, or Monte Carlo tree search); and (3) how to deal with multiple objectives. Moreover, the breadth-first and depth-first explorations are also unclear. The authors may write down the overall algorithm, which elucidates these explorations.

It is controversial whether Table 2 really suggests the superiority of the proposed method MatEvolve. On the one hand, MatEvolve does yield the best scores for the $S_{\text{val}}$ and $S_{\text{SSE}}$ metrics. On the other hand, $S_{\text{SSE}}$ is an aggregate of six metrics and MatEvolve performs worse than screening and LLMatDesign on half of them ($S_{\text{form}}$, $S_{\text{highT}}$, and $S_{\text{H}_2\text{O}}$). This showcases the danger of using one aggregate metric to replace multiple objectives when comparing methods.

Table 2 is only for the solid-state electrolytes use case. The authors should show a similar table for the other use case (cathode materials).

In Section 4.2.2, the authors state that "domain-specific knowledge plays a more decisive role than general coding capability" (in fact, this is mentioned multiple times in the paper). It is, however, unclear how the authors reach such a conclusion. Among the compared models, no one is known to be trained with more knowledge/data of the use case than are others. Furthermore, all models use the same curated knowledge base. How do the authors reach the conclusion that some models are better than others because they have more domain-specific knowledge?

**Questions:**

See the Weakness section above.

---

### Official Review · Reviewer_Ec9F · 2025-10-31

**Soundness:** 2
**Presentation:** 3
**Contribution:** 3
**Rating:** 4
**Confidence:** 3

**Summary:**

The paper presents MatEvolve, a symbolic–LLM agent for closed-loop, multi-objective materials design. It introduces (i) a Material Edit Language (MEL) for programmatic, Wyckoff-aware edits to crystal structures and stoichiometry; (ii) a curated Material Edit Base (MEB) distilled from literature into reusable MEL snippets; and (iii) a Material Evolution Engine (MEE) that runs a two-stage exploration (breadth-first → depth-first) with dynamic knowledge injection targeting the current performance bottleneck. Candidates are scored by a multi-objective fitness (MatScore) blending validity with battery-relevant surrogates (formation energy, electrochemical window, ionic conductivity, thermal/chemical stability for electrolytes; elasticity/stability/band gap/migration barriers for cathodes), e.g.
$S_{\text{SSE}}=\tfrac16\sum_{i\in{\text{form, elec, ion, highT, H2O, O2}}}\sigma(S_i)$ and
$S_{\text{Cathode}}=\tfrac14\sum_{i\in{\text{elastic, stab, gap, barrier}}}\sigma(S_i)$,
with $\sigma$ a sigmoid of normalized metrics. On solid-state electrolytes and LFP/LCO cathodes, MatEvolve reports higher composite scores than enumeration–screening and language-only baselines, with ablations attributing gains to MEL, dynamic knowledge injection, and the two-stage strategy.

**Strengths:**

Clear problem framing & architecture. Recasts materials design as programmatic evolution with a symbolic interface that is both chemically interpretable and LLM-friendly.

Practical knowledge integration. A curated MEB provides reusable edit patterns and is injected adaptively toward whichever metric is currently the bottleneck.

Balanced search strategy. The breadth-first → depth-first schedule plus adaptive focus on weak metrics is intuitive and effective.

Multi-objective fitness. MatScore aggregates task-relevant surrogates with explicit formulations for electrolytes vs. cathodes.

Ablations with attribution. MEL vs. formula-level edits, dynamic vs. static knowledge, and two-stage vs. single-stage exploration show clear contributions.

Modularity & portability. Same machinery extends from SSE to cathode design with consistent improvements.

**Weaknesses:**

Validation relies on surrogates. Conclusions depend heavily on proxy models; there is limited DFT or experimental spot-checking of top candidates.

Benchmark scope. Evaluations are centered on Li-based battery systems; generalization to other families (e.g., thermoelectrics, catalysts) is untested.

LLM comparison controls. Claims about smaller models outperforming larger ones may be confounded by decoding, prompt, and toolchain settings; stronger controls and variance reporting are needed.

MEB construction quality. Lacks quantitative assessment of extraction accuracy (precision/recall), inter-annotator agreement, or coverage of the literature space.

Sensitivity gaps. No systematic analysis of MatScore weighting, Wyckoff-based representation vs. alternatives, or the dynamic knowledge selector (Top-k, retrieval scoring).

MEL expressivity limits. Charge-compensation examples feel Li-centric; broader defect chemistry (anion/cation vacancies, heterovalent substitutions) may not be fully captured.

Reporting & reproducibility. Minor typos and limited details on wall-clock, token/cost accounting, hardware, seeds, and exact configs.

**Questions:**

1. Ground-truth validation: For top-ranked candidates, what fraction underwent DFT relaxation, and how often did MatScore trends hold post-relaxation? Any experimental checks?

2. MatScore calibration: How are per-metric normalizations and sigmoid scales chosen? Is there dynamic weighting, and how sensitive are results to these choices?

3. MEB quality: What are the precision/recall or error rates of MEL extraction? Any inter-annotator agreement metrics or held-out evaluation?

4. Generalization: Can you report at least one non-battery domain to test portability of MEL/MEB/MEE?

5. LLM controls: Can you repeat model-size comparisons with identical decoding/tool settings, multiple seeds, and variance bars?

6. Expressivity: Can MEL encode general charge balance and defect chemistry (anion/cation vacancies, heterovalent substitutions) beyond Li-specific patterns?

7. Retrieval sensitivity: How robust is dynamic knowledge injection to Top-k and retrieval model choices?

8. Compute profile: Please provide tokens/step, steps/experiment, total cost, and latencies for the end-to-end loop.

---

### Official Review · Reviewer_vBrG · 2025-11-01

**Soundness:** 1
**Presentation:** 1
**Contribution:** 1
**Rating:** 0
**Confidence:** 5

**Summary:**

The paper introduces a symbolic–LLM agent that performs material design as a closed-loop, programmatic evolution task. The agent uses a novel symbolic formalism that allows it to perform chemical operations programmatically.

**Strengths:**

Using LLM as a planner in sequential approach to material design is an attractive idea.

**Weaknesses:**

The narrative is too heavy on the boilerplates and PR-like verbiage.

The statement that the authors introduce a "novel closed-loop paradigm: insight-exploration-validation" indicates complete lack of awareness of the state-of-the-art in computational and experimental materials design.

The paper describes extremely small number of examples. It does not provide any meaningful evidence of general applicability and utility of the approach.

**Questions:**

This is beyond repair, unfortunately.

Please evaluate significantly more examples.

Please characterize trade-offs in exploration/exploitation regimes.

Please characterize the costs of validation of stability of novel materials.

Please characterize the approach to the assessment of synthetic viability of novel materials.

---

### Official Review · Reviewer_NdPX · 2025-11-01

**Soundness:** 2
**Presentation:** 3
**Contribution:** 2
**Rating:** 4
**Confidence:** 4

**Summary:**

This paper introduces MatEvolve, a framework for multi-objective materials design that combines a Large Language Model  with a symbolic system. The framework consists of three components: a Material Edit Language for programmatic modification, a Material Edit Base curated from scientific literature, and a Material Evolution Engine that guides the design process. The authors validate their approach on solid-state electrolyte and cathode design, showing improved performance over a screening baseline and another LLM-based method. The core finding is that domain-specific knowledge injection is more critical than LLM scale for such tasks.

**Strengths:**

Quality of Execution: The proposed system is well-designed and internally consistent. The development of the Material Edit Language and the systematic construction of the Material Edit Base from scientific literature represent a significant and high-quality engineering effort.

Clarity: The paper is very well-written and clearly structured. The figures and tables effectively illustrate the framework's architecture and experimental results, making the methodology easy to follow.

**Weaknesses:**

• Insufficient Experimental Comparison: The paper's primary weakness is the lack of comparison against state-of-the-art, domain-specific methods. Baselines are limited to a generic screening pipeline and another LLM-based approach. To substantiate claims of superiority, the framework must be benchmarked against established computational materials science methods like advanced genetic algorithms, Bayesian optimization, or specialized generative models (e.g., crystal diffusion models). Without this context, the reported performance gains are difficult to interpret.
•Limited Novelty of the Core Idea and Conclusion: The central methodology—augmenting an LLM with a domain-specific language and a curated knowledge base—is an established paradigm. Therefore, the main contribution of this work lies in successfully applying this paradigm to solve complex material optimization problems.

**Questions:**

1. The experimental validation is primarily against other LLM-based or generic methods. Why were established, domain-specific baselines from materials science literature (e.g., specialized evolutionary algorithms or surrogate-based Bayesian optimization) not included? How do you expect MatEvolve to perform against these highly optimized, non-LLM approaches?
2. The paper's conclusion about the importance of domain knowledge is a known principle. Are there unique challenges in materials science that make this finding particularly non-obvious or impactful compared to its application in other domains?
3. The construction of the MEB appears to be a manually intensive knowledge engineering process. How scalable is this approach? To what extent does the framework's success depend on this curation effort?

---

### Note · Authors · 2025-12-25

**Comment:**

We decided to withdraw this submission because we recently identified a technical error that requires significant time to correct. We plan to address this issue and resubmit in the future.

**Withdrawal Confirmation:**

I have read and agree with the venue's withdrawal policy on behalf of myself and my co-authors.